# Eye-Gaze Controlled Wheelchair Based on Deep Learning

**DOI:** 10.3390/s23136239

**Published:** 2023-07-07

**Authors:** Jun Xu, Zuning Huang, Liangyuan Liu, Xinghua Li, Kai Wei

**Affiliations:** 1School of Automation, Harbin University of Science and Technology, Harbin 150080, China; 2School of Electrical and Electronic Engineering, Harbin University of Science and Technology, Harbin 150080, China; 15268607136@163.com (Z.H.); 15938757827@163.com (L.L.); 15893334755@163.com (X.L.); 15765414898@163.com (K.W.)

**Keywords:** eye-tracking, deep learning, CBAM attention, wheelchair acceleration model

## Abstract

In this paper, we design a technologically intelligent wheelchair with eye-movement control for patients with ALS in a natural environment. The system consists of an electric wheelchair, a vision system, a two-dimensional robotic arm, and a main control system. The smart wheelchair obtains the eye image of the controller through a monocular camera and uses deep learning and an attention mechanism to calculate the eye-movement direction. In addition, starting from the relationship between the trajectory of the joystick and the wheelchair speed, we establish a motion acceleration model of the smart wheelchair, which reduces the sudden acceleration of the smart wheelchair during rapid motion and improves the smoothness of the motion of the smart wheelchair. The lightweight eye-movement recognition model is transplanted into an embedded AI controller. The test results show that the accuracy of eye-movement direction recognition is 98.49%, the wheelchair movement speed is up to 1 m/s, and the movement trajectory is smooth, without sudden changes.

## 1. Introduction

ALS is a progressive and fatal neurodegenerative disease that causes the degeneration of the patient’s upper and lower motor neurons, thereby weakening the muscles. Therefore, although many ALS patients are conscious, they cannot perform physical movements and verbal expressions. A growing body of research is dedicated to the application of artificial intelligence technologies to modify power wheelchairs to improve the quality of life of people with ALS. In the past few decades, researchers have carried out corresponding research on wheelchair motion control methods, including gesture control [1,2,3,4,5], voice control [6,7,8,9,10], eye-tracking control [11,12,13,14,15], and brain–computer interfaces [16,17,18,19,20,21,22]. These control methods can replace the rocker to complete the reading of the user’s motion direction intention and realize the motion control of the wheelchair. However, due to the loss of limb control and language communication abilities in patients with amyotrophic lateral sclerosis (ALS), gesture control and voice control are not viable options. In brain–computer interfaces, although the collection of brain signals largely eliminates noise interference, the semi-invasive or invasive electrodes used in these interfaces can pose risks to human health [23]. Compared with the above-mentioned control methods, eye-tracking control has unique advantages in terms of safety, portability, and practicality for patients with ALS.

Currently, research on the eye-tracking control of wheelchairs mainly focuses on two aspects: eye-tracking recognition and wheelchair control. Xiaokun Li et al. designed a head-mounted device based on an energy-controlled iterative curve-fitting method of infrared light, which can achieve precise pupil detection and tracking. The experimental results showed that the average tracking accuracy of the method for pupil rotation was at least 1.38% higher than that of conventional methods [24]. Fatma et al. proposed a method that uses a front camera to capture the user’s face information and determines the pupil center coordinates through a fuzzy logic controller to output wheelchair control decisions. This method features an adaptive threshold adjustment, enabling eye-tracking even under poor or unstable lighting conditions [25]. In terms of eye-tracking recognition, although the eye-tracker equipment has a high accuracy rate, the infrared light it generates can cause irreversible damage to the human eye, and it is far inferior to the camera recognition solution in terms of comfort and economic practicability.

In recent years, deep learning technology has been widely used in the fields of non-verbal communication, human–computer interaction, and intention prediction. Dliber et al. used the AlexNet network structure for eye-movement direction determination and trained the network model using a private dataset. The visualization results showed that the recognition accuracy of the model reached 97.88% [26]. Moayad Mokatren et al. designed a faster region-based convolutional neural network (RCNN) to detect pupils in infrared and RGB images. The experimental results showed that the method had a 3D gaze error of only 2.12° and could be used with high accuracy for various real-world moving scenes [27]. Ling Hu et al. proposed a model called SSD to detect eyeballs in images using a single deep neural network. The total average accuracy of target detection for this scheme was 91.85%, and the total average relative error after distortion removal was reduced to 1.25% [28]. The above-mentioned research based on deep learning provides a new direction for the development of eye-tracking technology.

For wheelchair control, Delia Dragusin et al. used a pupil-corneal reflex-based approach to accomplish sight estimation on a PC, and the wheelchair control chip received eye-movement commands from a PC via Bluetooth and controlled relay closures to drive motor motion [29]. Razvan Solea et al. used the PCI protocol to send the eye-tracking signal from a PC via a data acquisition card to a servo amplifier, which, in turn, sent a PWM control signal to the DC motor [30]. Aniwat Juhong et al. used x-axis–y-axis servos to control the wheelchair rocker so that the wheelchair could theoretically have a 360° movement direction, which greatly improved the maneuverability of the wheelchair [31]. Sho Higa et al. bypassed the joystick and directly sent the eye-tracking signal output via an embedded AI computing device to the control system of a WHILL CR electric wheelchair through a USB to complete the control of the wheelchair movement. Although this control method is simple and reliable, the electric wheelchair it is paired with is very expensive [32].

Deep learning-based eye-tracking methods heavily rely on the completeness and richness of the training data. For the public eye-gaze dataset, Wolfgang et al. compiled a dataset (TEyeD) from more than 20 million real-world eye images and pupil information, and the data were carefully labeled [33]. The MRL (Media Research Lab) built a large-scale dataset of human eye images with 15,000 pupil points (images) using different devices to capture drivers’ facial images under realistic conditions [34]. However, the existing dataset suffers from a lack of labeled samples, physical constraints on the recording process, and poor applicability to specific scenarios.

To sum up, to develop an intelligent eye-tracking wheelchair with low economic and operational barriers, here, we use a human eye-gaze model that combines convolutional neural networks and attention mechanisms in the eye-tracking recognition scheme. To further improve the generalizability and accuracy of the model, a large-range, multidimensional eye-tracking dataset is established. For wheelchair control, a structure that controls the movement of the wheelchair by manipulating the rocker with a tiller is developed to achieve precise control of the wheelchair. The intelligent eye-tracking wheelchair based on deep learning designed in this paper is an example of the application of artificial intelligence in medical care, elderly care, and other fields, and can serve as a reference for the use of emerging technologies to improve the quality of human life.

The rest of the paper is organized as follows. Section 2 presents a review of the related works. Section 3 outlines the intelligent eye-tracking wheelchair scheme developed in this paper and describes the acquisition process of the multidimensional eye-tracking dataset. Section 4 introduces the eye-tracking model-building and training methods. Section 5 describes the design of the eye-tracking wheelchair control system. Section 6 discusses the experiments and analyzes the results. Section 7 summarizes the article and suggests directions for future work.

## 2. Related Work

Eye-tracking methods can be divided into invasive and non-invasive methods. Invasive methods mainly include EEG (electroencephalogram) and EOG (electrooculogram) methods, whereas non-invasive methods can be further divided into appearance-based and model-based methods.

Intrusive methods use specific intrusive sensors to detect changes in the voltage, current, and magnetic field of eye movements to determine the direction of sight. The EOG method is an objective and quantitative retinal function detection method for detecting the electrostatic potential of the eye, which slowly changes with adaptation to light. To detect the horizontal movement of the eyeball, a pair of electrodes are placed on the inner and outer corners of the eye, respectively. To detect vertical movement, they are placed on the upper line of the eyelid, and the potential difference between the two is recorded. There is no potential difference when the eyeball is facing straight ahead, but when the eyeball is rotated, the electrode on the corneal side is more positive than the other electrodes [35,36]. The contact lens method involves implanting a contact lens consisting of reflective lenses into the human eye. The principle is to measure the reflection of different light rays by the reflective lens after different light rays enter the human eye, obtain the position of the center of the pupil by calculating the angle of the reflected light rays, and then judge the direction of sight. This method of directly embedding measuring equipment in the pupil is insensitive to changes in the external environment and has a high measurement accuracy; however, it is expensive, requires special equipment, and causes serious interference to users [37]. The above two methods determine the direction of the line-of-sight through a contact sensor device, although they are highly robust to changes in the external environment, such as lighting and head movement. However, in the process of use, the user needs to wear a special eye-contact device with large interference, which can have a significant impact on the comfort of the user and limit the degree of freedom of the user.

On the non-invasive side, many studies have applied model-based principles. Morimoto et al. proposed a sight estimation technique based on the pupil vector reflection method, which uses polynomial equations to describe the mapping relationship between the pupil corneal vector and the sight fall point. This method has high accuracy, but its head must remain absolutely still to interfere with the human degree of freedom [38]. Cerrolaza et al. used a camera and two infrared light-emitting tubes to calculate the relative distance between two spots formed by the corneal reflected infrared light, and then to calculate the mapping relationship between the pupil corneal reflection vector and the gaze point, and this method can solve the error caused by the small displacement of the head to some extent [39]. Hennessey et al. calculated the quadrilateral formed by four infrared reflected spots with a similar calibration. However, such methods require the passage of infrared light, and prolonged exposure to near-infrared light may cause damage to the eyes [40].

Appearance-based eye-tracking methods use image data as input and do not need to build 2D or 3D eye models to directly map image data to gaze points. In addition, the appearance-based approach [41,42] relies on the completeness and richness of the training data, i.e., the training data should contain all the situations present in the application scene, otherwise the approach cannot complete the mapping of image data to the gaze point. In the current application, a large number of image datasets are mainly used as the input to the neural network for training to achieve the mapping of image data to gaze points. The appearance-based method does not perform complicated operations such as camera and geometric scene calibration, which makes it less difficult to use and facilitates practical applications, without a complicated calibration process in the application.

Therefore, we designed a neural network model combining the convolutional neural network and attention mechanism for line-of-sight direction estimation from the aspects of economy, safety, practicality, and comfort of use.

## 3. Materials and Methods

### 3.1. Eye-Tracking Wheelchair Program

The eye-tracking wheelchair system consists of eye-tracking data acquisition, data preprocessing, eye-tracking direction estimation, and wheelchair motion control. The camera in front of the wheelchair transmits the face images collected in real time to the embedded AI computing module. The latter inputs the processed image into the neural network model to obtain the estimated direction of eye-tracking and transmits the signal to the Arduino; then, the Arduino controls the 2D servo to adjust the wheelchair rocker and change the motion state of the wheelchair. The user’s eye state and the estimated direction of eye-tracking are updated in real time on the display. The general block diagram of the eye-tracking wheelchair system is shown in Figure 1.

### 3.2. Dataset Creation

Datasets are the basis for training deep learning models, and the performance of deep learning models heavily depends on the quality and size of the datasets they are trained on. In order to further improve the practicability and accuracy of the model, 100 Chinese volunteers were recruited for this dataset collection. Through the two major scenes of virtual and reality, we recorded videos of volunteers gazing in different directions while completing tasks, extracted human eye images frame-by-frame using the OpenCv program and the Dlib algorithm, and automatically labeled the obtained data by the task attributes of the time period in which the frame was located.

#### 3.2.1. Multidimensional Eye-Tracking Data Acquisition

In this paper, we built the eye-tracking dataset through two dimensions: virtual and reality. Multidimensional datasets can capture the complex relationships between different features, better describe the characteristics and attributes of the data, provide a more comprehensive and accurate representation of the data, and enhance model performance.

(1)Virtual scene acquisition

In this paper, a virtual scene for eye-tracking direction detection was built under the robot operating system ROS using Gazebo software, as shown in Figure 2. The virtual scene is a maze, and there is a car in the maze, which matches the virtual camera so that the perspective of the volunteer is consistent with that of the car. The volunteers keep their heads still while watching the road in front of the car with their eyes and control the movement of the car through the keyboard, simulating the movement of a wheelchair in a real-life scenario. When the state of the car is shown (as in Figure 2), volunteers look at the feasible road on the right side of the wall, and at the same time control the car to the right through the keyboard. The volunteer’s facial image is captured and recorded by a camera placed in the center of the computer screen, while the volunteer’s actions on the keyboard are recorded by a script to achieve hand–eye synergy data collection. Through this method, the eye-tracking dataset in the virtual scene was established.

(2)Real scene acquisition

We set up an environment for eye-tracking data acquisition in a real scene, as shown in Figure 3b. The scene consisted of a wall with a nine-grid (as shown in Figure 3a), a laser pointer, and a wheelchair with a fixed camera. The nine-grid was a square area of 210 cm × 210 cm in size, each grid size was 70 cm × 70 cm, and a red sign was pasted in the center. When the volunteer’s eye gazes at the designated red sign, the direction of the human eye gaze becomes a nine-classification problem, which reduces the influence of the volunteer’s subjective behavior. In the real scene, the experimental assistant pointed to the red mark in the center of the grid with a laser pointer row-by-row from left to right, while the volunteer sat in a wheelchair and kept his head still, staring at the positions illuminated by the experimental assistant, and each position was maintained for 10 s. At the same time, the facial changes of the volunteers were recorded in real time by the camera on the wheelchair. Each frame of facial data was automatically marked according to the time the frame belonged to.

#### 3.2.2. Data Preprocessing

To ensure the accuracy of convolutional neural network-based algorithms, images need to be preprocessed before they are analyzed. The quality of the image has a direct impact on the accuracy of the algorithm. Different tasks require different image preprocessing methods to remove irrelevant information and enhance relevant information to improve task reliability. In this paper, the convolutional neural network was used to detect the human eye-tracking state, so redundant information other than human eyes needed to be removed in the preprocessing stage.

We chose to use the detector function of the Dlib library [43] to draw 68 feature points of the recognized face, used OpenCv to extract frames from the video, and used the face-detection algorithm of the Dlib library for each frame of the image. In order to effectively extract the irrelevant area information of the bridge of the nose in the two eyes, the images of the left and the right eyes were extracted, respectively, instead of directly extracting the images of both eyes. The comparison between the two is shown in Figure 4.

The area delineated by the feature points of serial numbers 42–47 is the corresponding area of the left eye, and the feature points of the corresponding area of the right eye are serial numbers 36–41. Taking the extraction of the left-eye image as an example, the minimum and maximum values of the abscissa and ordinate coordinates of the six points are the boundary area coordinates of the left-eye image. The following formula was used to obtain the boundary coordinates of the eye image. Among them, $x_{i},y_{i}$ are the corresponding abscissa and ordinate of each feature point, a,b are the minimum abscissa and ordinate of the eye area, and A, B are the maximum abscissa and ordinate of the eye area.
(1)$$\left\{\begin{array}{c} a=\underset{}{\min}\left(x_{i}\right)\\ b=\underset{}{\min}\left(y_{i}\right)\\ A=\underset{}{\max}\left(x_{i}\right)\\ B=\underset{}{\max}\left(y_{i}\right) \end{array}\right.$$

A total of 200 videos of volunteers staring in different directions in virtual and real scenes were saved. In order to extract the human eye information in each frame of the image, we processed the video frame-by-frame, and cut out the binocular pictures according to the feature point coordinates of human eyes. The part between the two eyes may have some noise and redundant information that interferes with the prediction of the gaze direction. For example, the nose bridge, eyeglass frames, eye shadows, and eye spacing between the two eyes may have an impact on the feature extraction of the eyes, thereby reducing the accuracy of estimating the gaze direction. By removing the part between the two eyes, the attention can be focused on the separate eye regions and the performance of the gaze direction recognition can be improved.

Considering the interference of the distance between the eyes on the human eye information, we intercepted the eye information of each frame of the picture, adjusted the size of the extracted picture to 100 × 50, and merged the eyes together, horizontally. According to the task attributes of the time of each frame of the image, the human eye information in each frame of the image was automatically marked, and finally, the collection of gaze data of 100 volunteers in virtual and real scenes was completed.

#### 3.2.3. Unification of Datasets

The data collected in the real environment in Section 3.2 had nine different labels, but after visualization by t-SNE [44], these labels could be directly transformed into three categories, which is also consistent with reality. For example, when the test volunteers gazed at the leftmost column, their eye features were approximately distributed in the same region regardless of the column they gazed at. Therefore, the three category labels could be cleverly used to uniformly mark the datasets of the real scene, which is exactly the same as the label classification of the datasets collected in the virtual environment, which is convenient for further dataset screening. The t-SNE visualization effect is shown in Figure 5.

For the convenience of description, we introduced some symbols: $X=\left\{X_{1},X_{2},\ldots ,X_{n}\right\}\;and\;n=1,2,\ldots ,100;Y=\left\{y_{1},y_{2},y_{3}\right\}.$ Among them, the X set represents 100 volunteers participating in the data collection. Each $X_{i}$ contains 1350 photos, and each photo has a corresponding label. Y represents the set of labels, $y_{1}$ corresponds to the label left, $y_{2}$ corresponds to the label forward, and $y_{3}$ corresponds to the label right. A total of 135,000 pictures with gaze labels in different directions were selected to provide training data for the algorithm training in the next section. The established dataset is shown in Figure 6.

## 4. Eye-Tracking Model Building

In this paper, we used deep learning for the estimation of the eye-gaze direction in the human eye-tracking recognition task for eye-tracking wheelchairs, which is divided into feature extraction networks and classification. This was completed by training a deep learning network (GazeNet) using the human eye database we have built, which incorporates several modules for feature extraction optimization, and tri-classifying the features for output using a cross-entropy loss function.

### 4.1. Eye-Gaze Direction Estimation

In this paper, we fully traded off the lightweight and accuracy of the algorithm when designing the network. In order to efficiently run the human gaze direction determination algorithm on embedded devices, we added the improved Inception module to reduce the model computation, and the ResBlock module and the CBAM attention module to improve the network performance in the designed GazeNet network structure, and the final overall network structure is shown in Figure 7. Each extracted human eye image first entered the improved Inception module after convolution and pooling operations, its output parameters entered the ResBlock module after convolution and pooling operations, and the resulting output entered the CBAM attention module after convolution and pooling operations. Then, the output features were again convolved and pooled before being tri-classified by the fully connected layer to achieve the eye-gaze direction estimation.

#### 4.1.1. Inception Module

The parallel structure adopted by the Inception module enables the input image to be processed by multiple convolutional kernels of different scales and pooling operations to obtain different levels of feature information [45]. The purpose is to extract features at different scales while keeping the size of the output feature map of the convolutional layer unchanged, to efficiently expand the depth and width of the network, and to prevent overfitting while improving the accuracy of the deep learning network. The specific implementation uses a combination of multiple convolutional kernels of different scales and pooling operations to increase the nonlinear representation of the model without increasing the number of model parameters. In this paper, we used the Inception module to decompose a 5 × 5 convolution kernel (Figure 8a) into two 3 × 3 convolution kernels (Figure 8b), so that we could effectively use only about (3×3+3×3)/(5×5)=72% of the computational overhead. This reduced the number of model parameters and computation while maintaining the same perceptual field, reducing the computational burden, as shown in Figure 8.

#### 4.1.2. ResBlock Module

Convolutional neural networks can extract a rich feature hierarchy, but there is also the hidden danger of gradient disappearance or gradient explosion, and the use of regularization processing may produce degradation problems. Therefore, we chose to add the ResBlock module in the middle layer of the network, which achieved a jump connection by directly adding the output and input of the convolutional layer, thus ensuring better gradient transfer during backpropagation and reducing the number of model parameters [46]. The $F\left(x\right)$ that needs to be taught can be written in the form of “residuals”, as follows:(2)$$F\left(x\right)=H\left(x\right)-x\;$$

During the training process, if the model finds that the gradient becomes very small when the neural network becomes deeper and deeper, i.e., the “degeneration phenomenon” appears, it can directly add the output and input to achieve a constant mapping [47]. The advantage of this is that even if the network is deep enough, it can guarantee effective feature extraction at each layer. In this way, the performance of the network can be maintained even if the network becomes deeper and deeper, avoiding the degradation problem that occurs in traditional neural networks, as shown in Figure 9.

X in Figure 9 represents the input, $H\left(X\right)$ represents the output obtained after a series of transformations in the network, and $F\left(X\right)$ represents the residual, which is the difference between the network output $H\left(X\right)$ and the input X. During the training process, the network automatically learns a set of appropriate weights, so that $F\left(X\right)$ converges to 0. The introduction of the residual concept allowed the network to better learn the mapping relationship between the input and the output, while reducing the risk of gradient disappearance or gradient explosion, thus improving the performance and generalization of the model.

#### 4.1.3. CBAM Module

In addition, since we wanted the feature processing to focus on the direction of the pupil in the eye, the Convolutional Block Attention Model (CBAM) was inserted at the back of the ResBlock module. The CBAM module can improve the performance of convolutional neural networks for eye-tracking direction estimation by better learning and representing specific image features through the attention mechanism [48]. The overall process of the CBAM module can be divided into two parts, as shown in Figure 10.

Channel attention is mainly used to capture the correlation between different channels [49]. This module first obtained two channels through global average pooling and maximum pooling operations: the 1×1×C channels. Then, it was mapped and activated by a two-layer neural network, with the number of neurons in the first layer being C/r. The number of neurons in the first layer is Relu, and the number of neurons in the second layer is C. A summary of the two obtained features was produced and passed through the Sigmoid activation function to obtain the weight coefficients, $M_{C}$, which was multiplied with the input feature mapping to obtain the output with enhanced channel feature representation. The features of an H × W × C input are F. The output feature formula is shown below:(3)$$\begin{array}{cc} M_{C}\left(F\right) & =\sigma \left(MLP\left(AvgPool\left(F\right)\right)+MLP\left(MaxPool\left(F\right)\right)\right)\\ & =\sigma \left(W_{1}\left(W_{0}\left(F_{avg}^{C}\right)\right)+W_{1}\left(W_{0}\left(F_{max}^{C}\right)\right)\right) \end{array}$$

In Equation (3), σ denotes the Sigmoid function, MLP denotes the multilayer perceptron, AvgPool denotes the average pooling layer, MaxPool denotes the maximum pooling layer, $W_{0}$ and $W_{1}$ denote the two weight matrices, $F_{avg}^{}$ and $F_{max}^{}$ denote the results of the input data after the AvgPool and MaxPool operations, and the superscript C is Channel, indicating the average pooling and maximum pooling operations in the channel dimension.

Spatial attention is mainly used to capture the correlation between different locations on the feature map [50]. This module first performed average pooling and maximum pooling to obtain two H×W×1 channels, respectively, which were stitched together and passed through a 7 × 7 convolutional layer with the Sigmoid activation function to obtain the weight coefficients, $M_{S}$. This weight was multiplied with the input feature mapping by means of element multiplication to obtain the output of the enhanced spatial feature representation [51]. An H×W×C input is characterized by F, and the output feature formula is shown below:(4)$$\begin{array}{cc} M_{S}\left(F\right) & =\sigma \left(f7\times 7\left(\left[AvgPool\left(F\right),MaxPool\left(F\right)\right]\right)\right)\\ & =\sigma \left(f7\times 7\left(\left[F_{avg}^{S};F_{max}^{S}\right]\right)\right) \end{array}\;$$

In Equation (4), σ denotes the Sigmoid function, AvgPool denotes the average pooling layer, MaxPool denotes the maximum pooling layer, W_0_ and W_1_ denote two weight matrices, f7×7 denotes a convolution kernel of size 7 × 7, $F_{avg}^{}$ and $F_{max}^{}$ denote the results of the input data after the AvgPool and MaxPool operations, and the superscript S is Spatial, indicating the average pooling and maximum pooling operations in the spatial dimension.

### 4.2. Classification Network

#### 4.2.1. Fully Connected Layer

The classification estimation task was implemented by a fully connected layer integrating the local features after the convolution operation through a weight matrix. In the eye-tracking direction classification task of this paper, it was necessary to obtain the result of determining whether the eye is forward, left, or right, and this, therefore, is a triple-classification problem. The output of the feature extraction network in this paper was 2×2×4, and the fully connected layer had 3 neurons. First, the output of the feature extraction network was flattened into a one-dimensional column vector: $x={\left[x_{1},x_{2},\cdots ,x_{15},x_{16}\right]}^{T}$; then, for each neuron in the fully connected layer: $Z={\left[Z_{1},Z_{2},Z_{3}\right]}^{T}$, a linear operation was performed with each element in x. In the forward-propagation process, the fully connected layer can be viewed as a linear weighted summation process. Specifically, each node in the previous layer was multiplied by a weighting factor, w, and a bias, b, was added to obtain the corresponding output on the fully connected layer, z. The computational process of the fully connected layer can be expressed using Equation (5):(5)$$\left[\begin{array}{c} z_{1}\\ z_{2}\\ z_{3} \end{array}\right]=\left[\begin{array}{cc} \begin{array}{c} w_{11}\\ w_{21} \end{array} & \begin{array}{c} \begin{array}{ccc} w_{12} & \cdots & w_{16} \end{array}\\ \begin{array}{ccc} w_{22} & \cdots & w_{26} \end{array} \end{array}\\ w_{31} & \begin{array}{ccc} w_{32} & \cdots & w_{36} \end{array} \end{array}\right]*\left[\begin{array}{c} x_{1}\\ \begin{array}{c} x_{2}\\ \vdots \end{array}\\ \begin{array}{c} x_{15}\\ x_{16} \end{array} \end{array}\right]+\left[\begin{array}{c} b_{1}\\ b_{2}\\ b_{3} \end{array}\right]$$

The output was normalized using the SoftMax function, which maps a vector into a probability distribution, where each element is a non-negative number, and the sum of all elements is 1. Thus, for a multiclassification problem with n classes, the SoftMax function can convert an n-dimensional vector into a probability distribution, where each element represents the predicted probability of that class. The neuron $Z={\left[Z_{1},Z_{2},Z_{3}\right]}^{T}$ on the fully connected layer was normalized to $y={\left[y_{1},y_{2},y_{3}\right]}^{T}$ by the SoftMax function with the constraint: $y_{1}+y_{2}+y_{3}=1$. The transformation relation between $y_{i}$ and $Z_{j}$ is:(6)$$y_{i}=\frac{e^{Z_{j}}}{\sum_{j=1}^{3}e^{Z_{j}}}$$

#### 4.2.2. Cross-Entropy Loss Function

During the training process, we wanted to make the output probability distribution of the model as close as possible to the true probability distribution, so we needed to design a suitable loss function to measure the difference between them. The cross-entropy function can effectively measure the difference between two probability distributions and is, therefore, widely used in classification problems [52]. In particular, if for a sample x with a true label of y, the predicted output of the SoftMax model for it is $\hat{y}$, then the cross-entropy loss function can be expressed as:(7)$$Loss\left(y,\hat{y}\right)=-\sum_{i}y_{i}\ln\hat{y}$$

This loss function can be regarded as the KL dispersion (Kullback–Leibler divergence) between the true and predicted labels [53]. The cross-entropy loss function obtained the minimum value of 0 when the output probability distribution of the model was exactly the same as the true probability distribution; however, when the difference between them increased, the value of the cross-entropy loss function also increased. Since the SoftMax function mapped the $y_{i}$ values to between 0 and 1, and according to the constraint $\sum_{i}y_{i}=1,$ it can be deduced that:(8)$$\frac{e^{z_{i}}}{\sum_{j=1}^{3}e^{z_{j}}}=1-\frac{\sum_{j\neq i}e^{z_{j}}}{\sum_{j=1}^{3}e^{z_{j}}}$$
when $y_{i}=1$, the loss function is:(9)$${Loss}_{i}\left(y,\hat{y}\right)=-\sum_{i}\ln y_{i}$$

The derivative procedure for ${Loss}_{i}\left(y,\hat{y}\right)$ is as follows:(10)$$\begin{array}{cc} \frac{\partial {Loss}_{i}}{\partial z_{i}} & =-\frac{\partial \ln y_{i}}{\partial z_{i}}=\frac{\partial \left(-\ln\frac{e^{z_{i}}}{\sum_{j}e^{z_{j}}}\right)}{\partial z_{i}}=-\frac{1}{\frac{e^{z_{i}}}{\sum_{j}e^{z_{j}}}}\cdot \frac{\partial \left(\frac{e^{z_{i}}}{\sum_{j}e^{z_{j}}}\right)}{\partial z_{i}}\\ & =\frac{\sum_{j}e^{z_{j}}\cdot \sum_{j\neq i}e^{z_{j}}}{e^{z_{i}}}\cdot \frac{-e^{z_{i}}}{{\left(\sum_{j}e^{z_{j}}\right)}^{2}}=-\frac{\sum_{j\neq i}e^{z_{j}}}{\sum_{j}e^{z_{j}}}\\ & =y_{i}-1 \end{array}$$

From the derivation of the formula, it can be seen that the clever use of the cross-entropy loss function with SoftMax for the triple-classification task makes it very easy to calculate the gradient in the backpropagation. The gradient of the backward update was obtained by simply taking the $y_{i}-1$ calculated in the forward direction.

## 5. The Design of the Eye-Tracking Wheelchair Control System

In the previous section, the eye-tracking recognition algorithm was investigated. A complete eye-tracking wheelchair control system design solution should also include data acquisition, data processing, motion control, human–machine interaction, and system optimization. The physical diagram and hardware data diagram of the wheelchair designed in this paper are shown in Figure 11.

According to the role of hardware in the system, it can be divided into three sections: the data acquisition section, the data processing section, and the motion control section. The data acquisition area consists of the Gook HD98 HD camera with a resolution of 1920 × 1080 and a 10-inch touchscreen. The camera is responsible for capturing face images and the touch screen displays eye-tracking information in real time. The data processing section consists of the Jeston Tx2, which has 256 CUDA cores and up to 8 GB of memory. Its arithmetic power is comparable to that of a desktop-class graphics card GTX750, and it can easily handle the computational task of eye-tracking recognition. The motion control section is composed of Arduino as well as MG995 servo. Arduino receives an eye-tracking signal and drives MG995 to control the rocker to change the wheelchair motion.

Common power wheelchair modifications are adjustments to the hardware portion of an existing wheelchair. Changes to the wheelchair motion can be achieved by simply controlling the rocker during the wheelchair motion. Therefore, in this paper, a mechanical structure (as shown in Figure 12) was installed on the wheelchair rocker controller. The structure consists of a base, a servo bracket, a control rod extension, and a control arm. The two servos are the x-axis and y-axis servos, which cooperate with each other to realize the control of the electric wheelchair rocker by turning in different directions according to the control command, so as to fulfill the control of the direction of wheelchair movement.

In the upper computer TX2, we designed the intelligent eye-tracking wheelchair control platform through PYQT5. The intelligent eye-tracking wheelchair control platform contains a data acquisition section and a wheelchair control section. The data acquisition section collects face and human eye images in real time. The wheelchair control section contains a wheelchair start button, wheelchair motion direction display, camera selection, and face count display functions. The intelligent eye-tracking wheelchair control platform makes it easier to achieve wheelchair activation and monitor wheelchair movement. The interactive interface is shown in Figure 13.

### 5.1. Motion Control Optimization

In the process of wheelchair steering, the motion path of the rocker will have an important impact on the acceleration change in the system because of the non-linear relationship between the left and right wheel speeds and the rocker angle, as shown in Figure 14. In order to make the wheelchair motion change smoothly and weaken the risk of vibration and even rollover caused by the sudden change of acceleration, this section presents the research on the rocker trajectory-tracking control.

In order to establish the rocker control model, we introduced a polar coordinate system in the rocker motion plane, specifying the distance from the rocker to the reset as r, and the angle between the projection line of the rocker on the plane and the positive left as θ. The coordinate diagram is shown in Figure 15.

Since the speed of the left and right wheels is symmetrical with respect to the change in the rocker angle, and the speed of the right wheel is almost constant in the interval of $0^{{}^{\circ}}-90^{{}^{\circ}}$, therefore, it is only necessary to analyze the velocity change curve of the left wheel at $0^{{}^{\circ}}-90^{{}^{\circ}}$. The acceleration, a, of the left wheel is related to the velocity, $V_{w}$, and the angle, θ, of the rocker in polar coordinates, as follows:(11)$$a=\frac{dV_{w}}{d\theta }\cdot \frac{d\theta }{dt}$$

In the range of $0^{{}^{\circ}}-90^{{}^{\circ}}$, $\frac{dV_{w}}{d\theta }$ continuously decreased as θ increased. In order to make the a change more smoothly, it was necessary to make $\frac{d\theta }{dt}$ continuously increase as θ increased. Accordingly, the following θ−t diagram was drawn, as in Figure 16.

By:(12)$$\left\{\begin{array}{c} \theta (t)=\int_{0}^{t}\omega \left(t\right)dt\\ \omega (t)=\frac{V_{R}}{r(t)} \end{array}\right.$$

It was obtained as follows:(13)$$r\left(t\right)=\frac{V_{R}}{\theta^{\prime }\left(t\right)}$$
where ω(t) is the rocker angular velocity and $V_{R}$ is the rocker velocity, which can be considered constant in magnitude. Then, r(t) could be obtained according to Equation (13) with the derived θ(t), and the rocker motion trajectory could be determined accordingly.

To simplify the analysis, $V_{w}(\theta )$ as well as θ(t) can be expressed as Equations (14) and (15):(14)$$V_{w}\left(\theta \right)=a_{1}\theta^{2}+b_{1}\theta \left(\theta <90^{{}^{\circ}}\right)$$
(15)$$\theta \left(t\right)=a_{2}t^{2}+b_{2}t\left(t<T_{e}\right)$$
where $a_{1}$, $b_{1}$, and $T_{e}$ are known, $T_{e}$ denotes the time required for the rocker to move from the starting position to the end position, and $\theta \left(T_{e}\right)=\frac{\pi }{2}$.

Then, substituting Equations (14) and (15) into (11) yields:(16)$$a\left(t\right)=\left[2a_{1}\left(a_{2}t^{2}+b_{1}t\right)+b_{1}\right]\left(2a_{2}t+b_{2}\right)$$

The simplification yields:(17)$$a\left(t\right)=A\left(t-t_{1}\right)\left(t-t_{2}\right)\left(t-t_{3}\right)$$
with the following constraints:(18)$$\left\{\begin{array}{c} a\left(0\right)=0\\ a\left(T_{e}\right)=0\\ \int_{0}^{T_{e}}a\left(t\right)=V_{e} \end{array}\right.$$

Substituting Equation (17) into (18) yields:(19)$$a\left(t\right)=\frac{12V_{e}}{2t_{3}{T_{e}}^{3}-T_{e}^{4}}t\left(t-T_{e}\right)\left(t-t_{3}\right)$$
where $T_{e}$ indicates the time taken for the rocker to move from $0^{{}^{\circ}}$ to $90^{{}^{\circ}}$. According to the characteristics of the cubic function, it is necessary to satisfy $t_{3}>T_{e}$ in order to make the acceleration change more smoothly in $0-T_{e}$. Then, Matlab can be used to plot the curve of a with time t when $t_{3}$ changes.

From Figure 17, it can be seen that the curve of $t_{3}$ with t tended to flatten out during the increase of $t_{3}$ from $T_{e}$, and the maximum value tended to be $\frac{1.5V_{e}}{T_{e}}$. Accordingly, θ(t) and r(t) under the target conditions could be derived, and the rocker motion trajectory could then be derived from them.

### 5.2. System Flow

#### 5.2.1. Blink Detection

The face-detection algorithm using the Dlib library with OpenCV image processing to obtain periocular feature point data was described in Section 3.2.2, and the eye-tracking estimation model was built in Section 3. Taking the left eye as an example, the outline of the eye was first located in the eye image, and points p_1_–p_6_ were used to represent the key points of the eye, as shown in Figure 18. The actual use of the system control needs to detect whether the human eye is open before the line-of-sight estimation. The eye opening and closing states can be determined by calculating the eye aspect ratio (EAR) [54] in real time and comparing it with the set threshold, and there is no need to enter the line-of-sight estimation procedure if the eye opening and closing do not meet the EAR threshold.

The respective EAR values of the left and right eyes were calculated, and finally, the average EAR value of the left and right eyes was obtained. When the eyes were open, EAR maintained a constant value; when the eyes were closed, the EAR value tended to 0. If the calculated EAR value was less than the set threshold, it was determined to be a blink action. The EAR calculation formula is shown below:(20)$$EAR_{left}=\frac{\left\|p_{2}-p_{6}\right\|+\left\|p_{3}-p_{5}\right\|}{2\left\|p_{1}-p_{4}\right\|}$$
(21)$$EAR=\frac{EAR_{left}+EAR_{right}}{2}$$

#### 5.2.2. Saccades Processing

In addition to the impact that blinking can have on wheelchair control, the sweeping behavior of the user due to unexpected events during use may also affect the control of the wheelchair and the safety of the user. If the user rapidly changes the direction of gaze in a short period of time, the wheelchair may continuously receive movement commands in different directions, causing the wheelchair to swing from side to side. To reduce the risk of this unexpected situation, we used the results that occurred four or more times out of every eight classification results as the motion control command according to the majority principle, and if there was no majority result out of eight classification results, the wheelchair motion stopped. Since the network model is capable of generating approximately 16 classification results in 1 s, the above treatment had less impact on wheelchair maneuverability.

#### 5.2.3. System Flow Chart

After analyzing the workflow of the eye-tracking wheelchair in a previous paper, the system flowchart of the eye-tracking wheelchair is presented in this paper, as shown in Figure 19. The control program begins in the host computer, and the camera in front of the wheelchair is automatically turned on by the program, and then the camera starts to extract facial features and collect eye-movement data. Jeston Tx2 calculates the EAR of each frame of the acquired eye-movement data. If the latter does not exceed the threshold, the eye-movement data are reacquired, and if the threshold is exceeded, the frame is fed into the GazeNet network model to calculate the eye-tracking direction and the classification result is output to the motion controller, Arduino. The latter manipulates the rocker to complete the control of the wheelchair movement based on the eye-movement signal. If the wheelchair is turned off at this point, the system stops working; otherwise, it will return to the eye-movement data acquisition process and continue to complete the calculation and judgment of the eye-movement data EAR, and so on.

## 6. Experiments and Results Analysis

In the previous sections, we completed the establishment of the model and the construction of the system. In order to verify the practicality and reliability of the eye-tracking wheelchair, in this section, we outline the test experiments conducted on the accuracy of the model recognition results and the accuracy of the wheelchair control. The first experiment was a comparative experimental evaluation of the GazeNet proposed in this paper with three existing models, such as AlexNet, ResNet18, and MobileNet-V2, to demonstrate the superiority of the model used in this paper in eye-tracking recognition. The second experiment quantified the wheelchair control accuracy by testing the deviation of the actual wheelchair motion trajectory from the target path, and the third experiment tested the effect of motion control optimization in Section 4.2 by measuring the Arduino output PWM signal.

### 6.1. Evaluation of GazeNet’s Effectiveness

#### 6.1.1. Hyperparameter Optimization

In the training and testing of the GazeNet network, each image in $\bar{X}$ was subjected to the preprocessing operation described in Section 3.2.2 to obtain $\bar{X}$, which was named the multi-environment attentional gaze dataset. The 135,000 binocular images in $\bar{X}$ were divided into a training set, validation set, and test set in the ratio of 98:1:1 and put into the network for training. The initial parameters were set with random initialization, the learning rate was 0.001, the optimizer used SGD (stochastic gradient descent) [55], the loss function used cross-entropy, and accuracy was used as an important evaluation index to measure the model performance. The accuracy rate was calculated as follows:(22)$$Accuracy=\frac{TP+TN}{TP+TN+FP+FN}$$
where TP is true positive, TN is true negative, FP is false positive, and FN is false negative.

As shown in Figure 20, a total of 30 rounds were trained, and after 18 epochs of training, the accuracy curve gradually leveled off and no longer significantly improved, which indicated that the training of the classification network was completed. In this paper, we chose to save the weight parameters of the model at the 29th round, and its accuracy rate was 0.98494.

#### 6.1.2. Assessment Measures and Methods

The GazeNet network described above was trained on the MEAGaze dataset using the same training set as the widely used AlexNet [56], ResNet18 [57], and MobileNet-V2 [58] models, and tested under the same test set. The test results are shown in Table 1 and Figure 21. The GazeNet network in this paper achieved an accuracy of 98.49% and used the smallest number of parameters, only 125,749, which was significantly better than the other three models. The comparison showed that the GazeNet network was about 1% point more accurate than the ResNet18 model, with the second-highest accuracy rate, while the former had only 22.3% of the number of participants of the AlexNet model, with the second-lowest parameter amount. The GazeNet network has a minimum number of parameters with excellent accuracy, which makes it an ideal lightweight model that can meet the needs of both high accuracy and low resource consumption.

### 6.2. Reliability Analysis of Eye-Tracking Wheelchair Control

The comparative experiments in the previous section demonstrated the high recognition rate of the eye-tracking model used in this paper. In order to verify the reliability and control accuracy of the eye-tracking wheelchair, we launched experiments related to the eye-tracking wheelchair in specific scenarios.

The experimental route diagram is shown in Figure 22. The solid part represents the target path of the wheelchair center, and the dashed part is the target path of both wheels. Since the width of the wheelchair that we used was 60 cm, the dashed path spacing was 60 cm. An experiment designed in this paper consisted of two laps of the line: the yellow line in Figure 22a is the target path of the first lap, the blue line in Figure 22c is the target path of the second lap, and Figure 22b is the real target line diagram combining the two, where the green line represents the overlapping part of the two laps of the line. The start and end points of both laps are the red points in Figure 22. The experimental design allowed the wheelchair to complete an experiment with an equal number of left and right turns, both eight times, making the experimental data more meaningful.

To make the data of the experiment generalizable, we recruited 20 volunteers aged 20–35 years to participate in the experiment. These 20 volunteers were 50% men and 50% women and wore glasses. Each volunteer was trained to operate a wheelchair while maintaining head immobilization to complete the experiment, during which the speed of the wheelchair was set to 1 m/s. We installed a laser distance measurement module on each side of the wheelchair, as shown in Figure 23, which had a range of up to 80 m, a measurement accuracy of 1.0 mm, and a measurement frequency of 20 Hz. It can measure and record the distance to the wall in real time and obtain the deviation of the wheelchair center from the target path based on the distance to the wall. The values of deviation with a distance of movement at each experiment were totaled and divided by the total number of people, and a graph of deviation with a distance of movement was obtained, as shown in Figure 24. The fluctuating part of this figure is the deviation of the wheelchair during the turn, and the maximum deviation appeared at the second circle of 8 m, which was 6.76 cm. It can be seen that the deviation during the wheelchair movement was small, the accuracy of the model and control was high, and the eye-activated wheelchair we designed had high practicality.

In Section 4.2, we optimized the rocker control. To verify the effect of this work on the stability of the wheelchair control, we had the experimenter sit on the motorized wheelchair, disconnect the motor power, and connect the Arduino output pins with an oscilloscope. The experimenter simulated the eye-movement state in the previous experiment while keeping the head immobile. The average curve of the rudder duty cycle over time during the experiment was obtained by superimposing the numerical curves of the rudder duty cycle over time during the experiment for the 20 volunteers and applying the total number of people, as shown in Figure 25.

As can be seen from Figure 25, the control of the servo was smoother and without noise, which can also indicate the high accuracy of the neural network triple classification and the absence of several types of jumping outputs.

## 7. Conclusions

We collected an eye-movement dataset with 135,000 annotated images through virtual and real scenes and proposed a GazeNet eye-movement neural network model based on the three-category dataset. The model comparison experiment results showed that the GazeNet model proposed in this paper had a faster convergence speed and higher accuracy than the other three models. In terms of wheelchair control, we used a 2D steering gear to control the joystick and optimize the steering gear control signal. The follow-up Arduino output waveform experiment showed that the steering gear control signal was smooth and gentle, and the optimization results were good. In addition, the experiment also proved that the three-category result of the model was accurate, and there was no case of jumping outputs of several eye-movement categories. In the wheelchair control reliability analysis, we obtained the deviation between the target and the actual trajectory during the movement process through laser ranging and concluded that the accuracy of the motion control and eye-movement model was high.

However, the current motion control part is relatively complicated, and the upper limit of optimization is low. If the signal of Jeston Tx2 can be directly output to control the wheelchair, the motion control can be made simpler and more reliable. In addition, the eye-movement dataset was also collected in a specific scene and did not fully consider applications in scenes in daily life, such as crossing the road, rainy days, etc. After the experiment, many volunteers reported that continuous eye-movement control increased the burden on users, which reduced the accuracy of control to a certain extent and increased the risk for users. Considering that the current wheelchair control scheme lacks adaptability to the environment, we expect to add a visual slam and path planning [59] to the control part in the follow-up work. During the process, the direction of sight can be freely controlled, and the wheelchair can autonomously perceive the surrounding environment and make adjustments to reach the destination. For safety reasons, we plan to add a positioning system to the wheelchair. When the user operates the wheelchair, their family members can remotely know the user’s specific location through the mobile phone program.

## Figures and Tables

**Figure 1 sensors-23-06239-f001:**
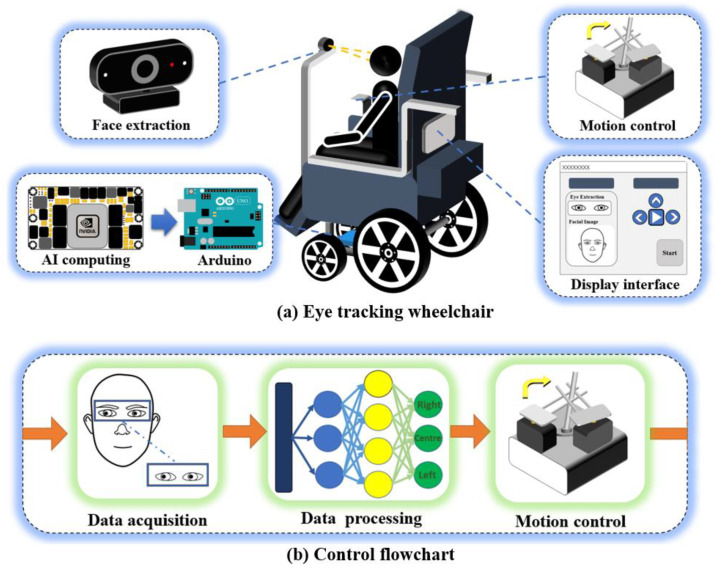
General block diagram of intelligent eye-tracking wheelchair system.

**Figure 2 sensors-23-06239-f002:**
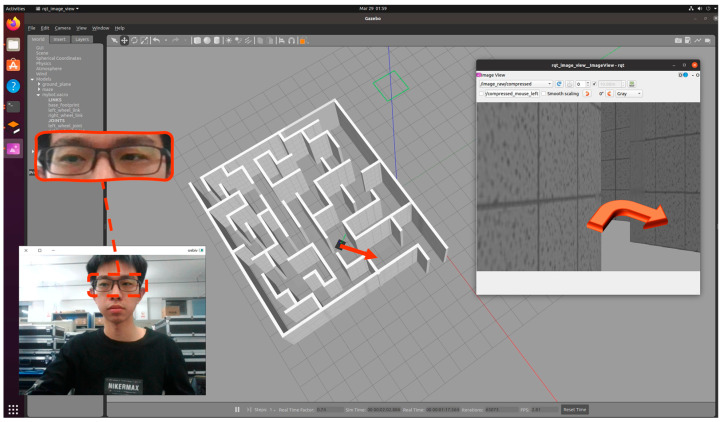
Virtual eye-tracking data acquisition.

**Figure 3 sensors-23-06239-f003:**
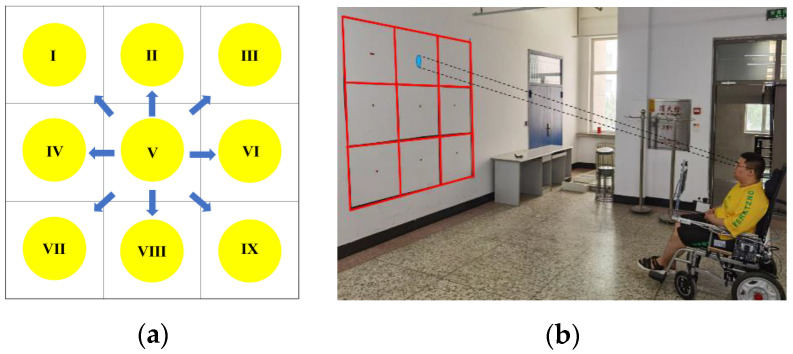
Real eye-tracking data acquisition: (**a**) nine-grid and (**b**) real scene.

**Figure 4 sensors-23-06239-f004:**
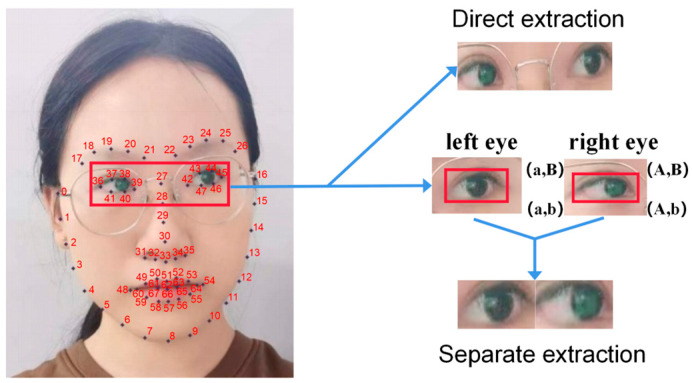
Collecting Dlib key points to extract human eye pictures.

**Figure 5 sensors-23-06239-f005:**
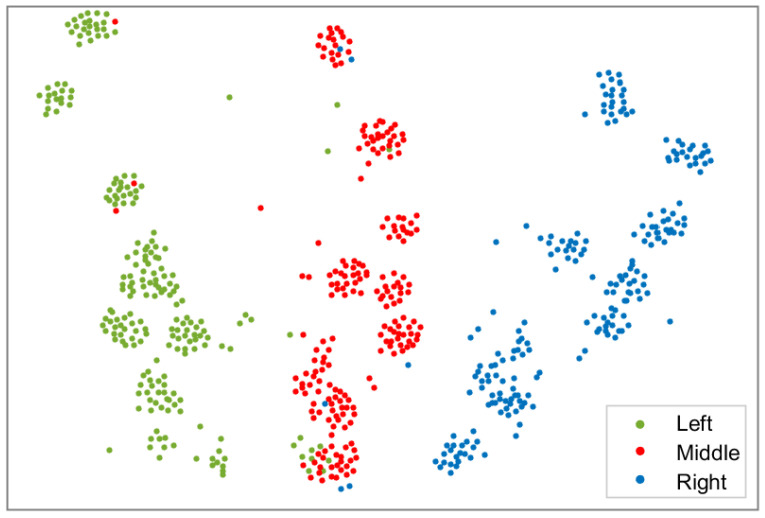
t-SNE visualization effect diagram.

**Figure 6 sensors-23-06239-f006:**
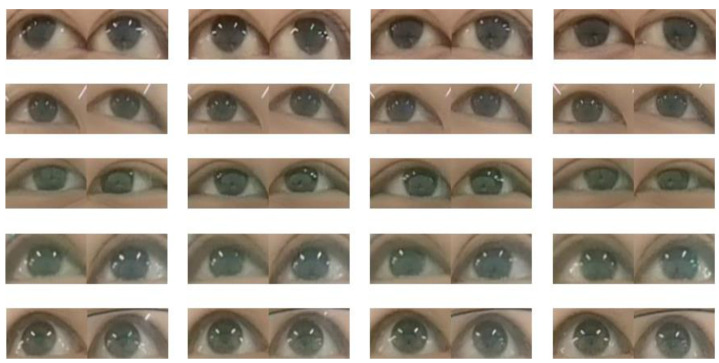
Eye-tracking dataset.

**Figure 7 sensors-23-06239-f007:**
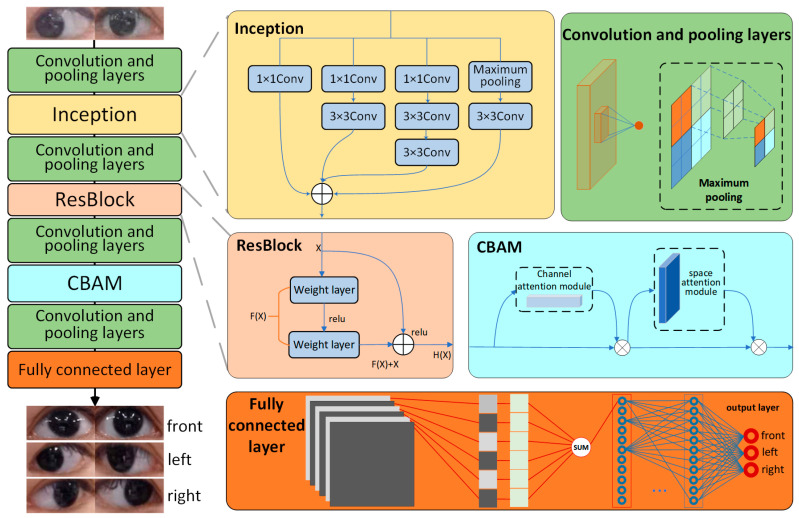
GazeNet structure diagram.

**Figure 8 sensors-23-06239-f008:**
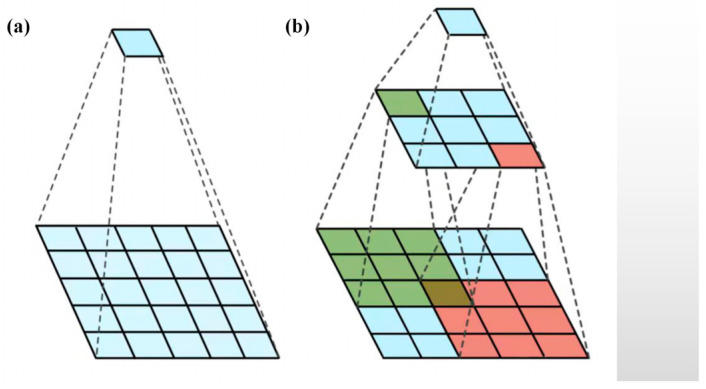
Sensory fields corresponding to two different convolution operations: (**a**) 5 × 5 convolution kernel and (**b**) 3 × 3 convolution kernel.

**Figure 9 sensors-23-06239-f009:**
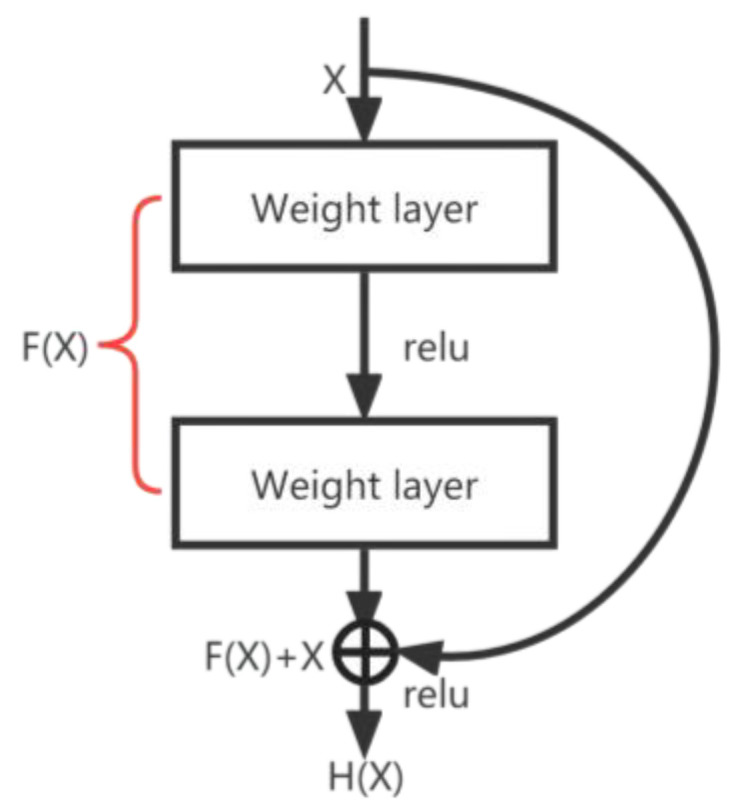
ResBlock module.

**Figure 10 sensors-23-06239-f010:**
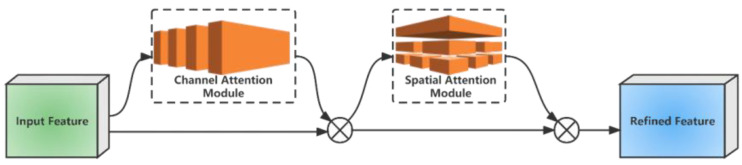
Convolutional Block Attention Model.

**Figure 11 sensors-23-06239-f011:**
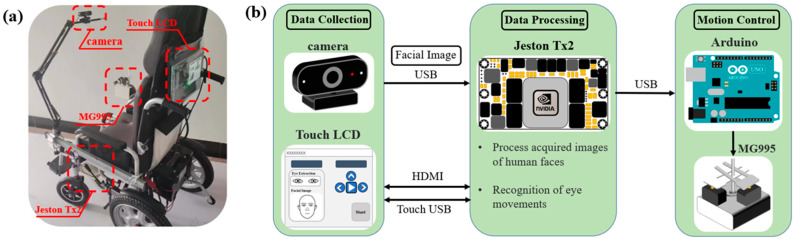
Hardware design diagram: (**a**) wheelchair physical picture and (**b**) hardware data flow diagram.

**Figure 12 sensors-23-06239-f012:**
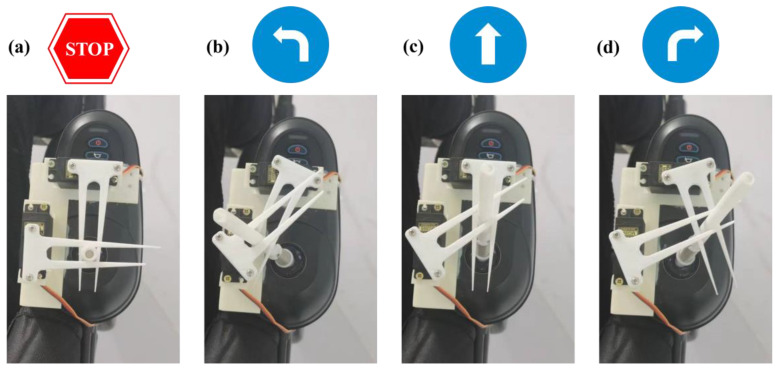
Modified wheelchair controller: (**a**) stop, (**b**) left, (**c**) forward, and (**d**) right.

**Figure 13 sensors-23-06239-f013:**
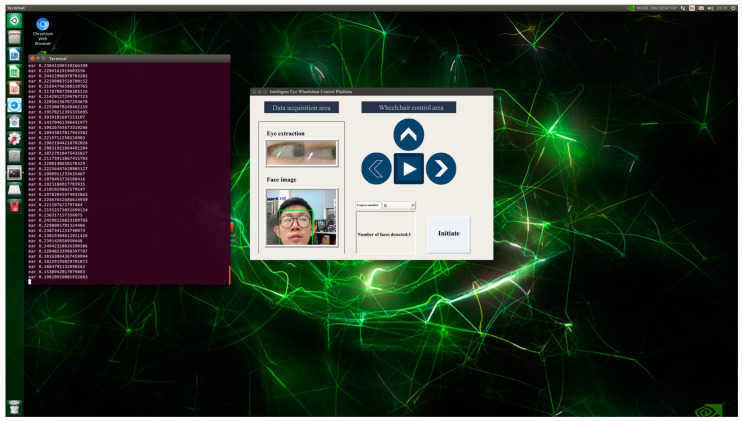
Interactive interface.

**Figure 14 sensors-23-06239-f014:**
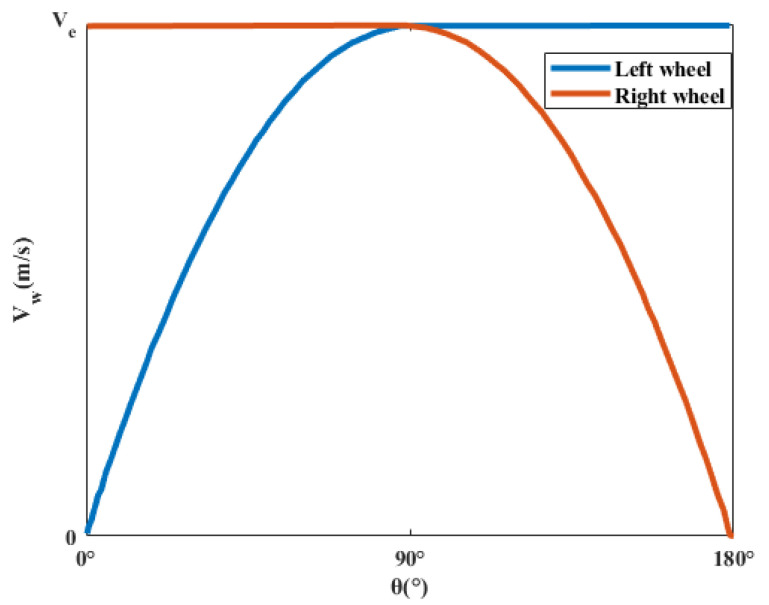
Fitting diagram of wheelchair speed and rocker angle.

**Figure 15 sensors-23-06239-f015:**
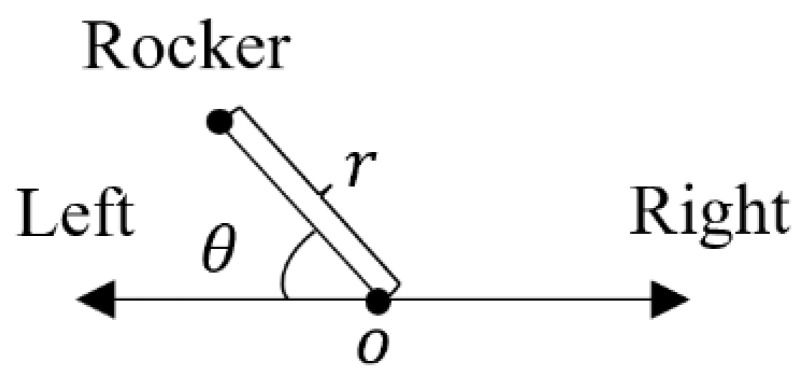
Polar coordinate diagram of rocker position.

**Figure 16 sensors-23-06239-f016:**
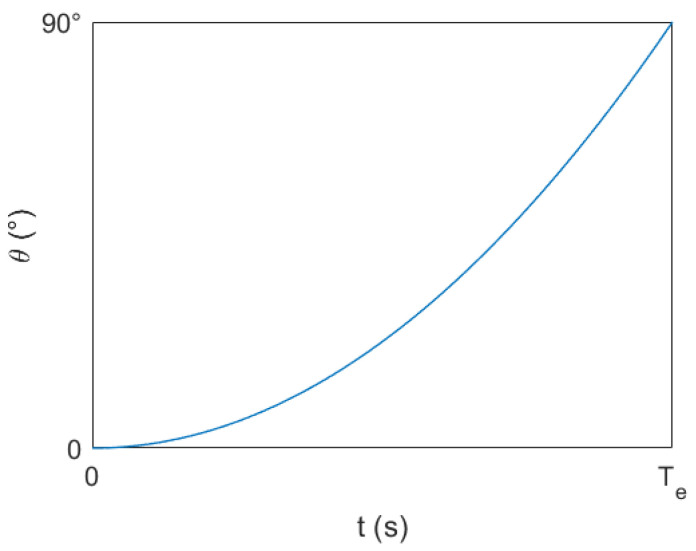
*θ*-*t* diagram.

**Figure 17 sensors-23-06239-f017:**
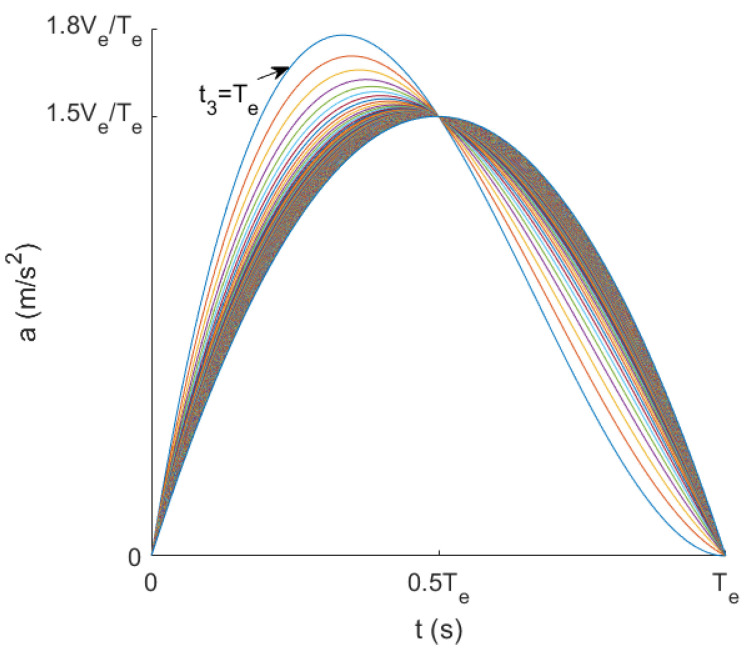
Variation curve of a with t at different $t_{3}$.

**Figure 18 sensors-23-06239-f018:**
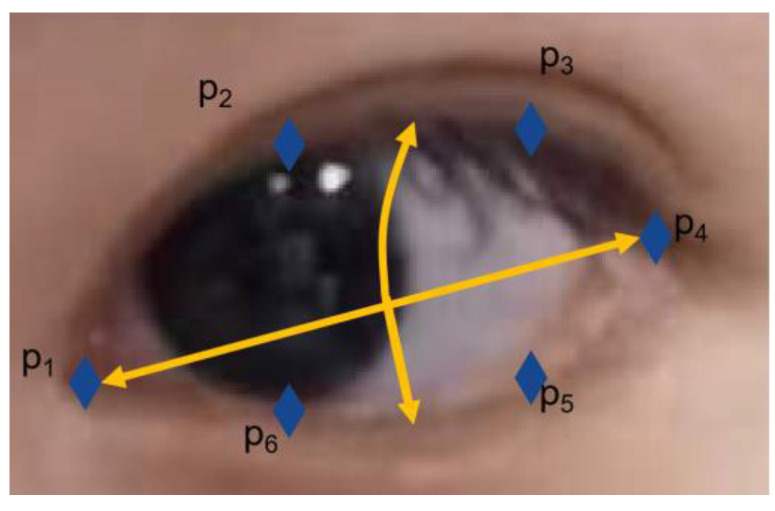
Key points of the eye.

**Figure 19 sensors-23-06239-f019:**
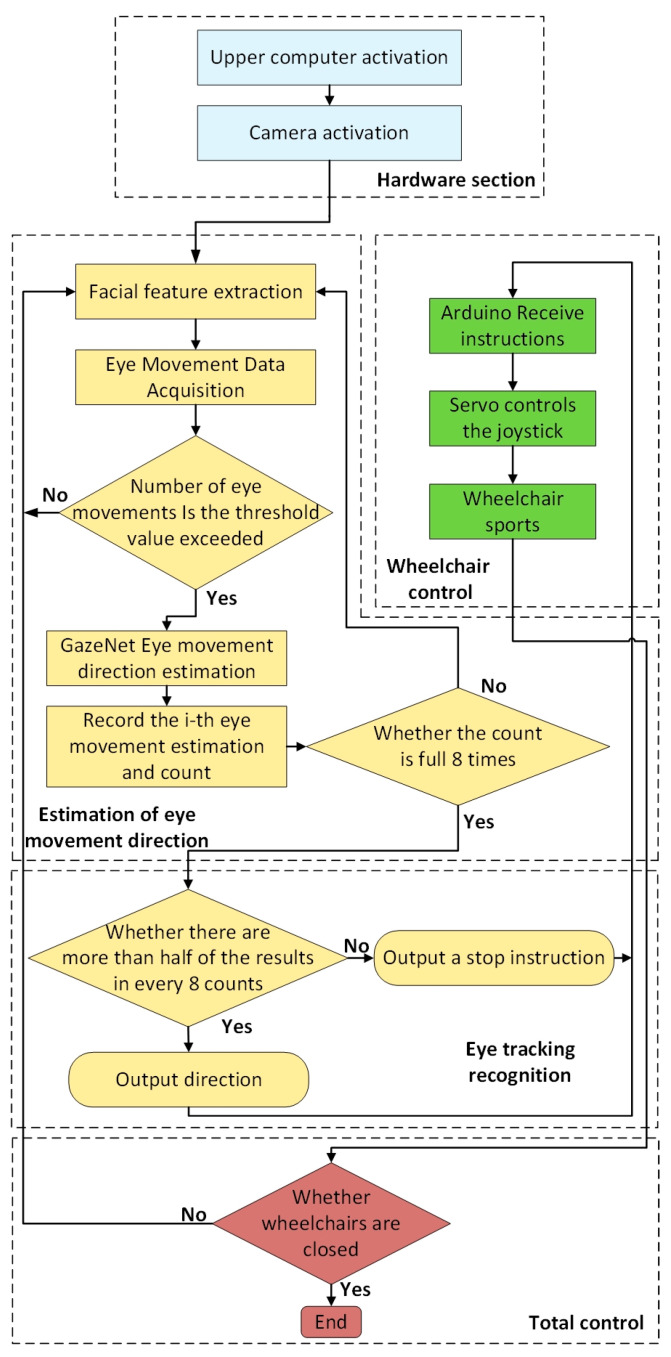
System flow chart.

**Figure 20 sensors-23-06239-f020:**
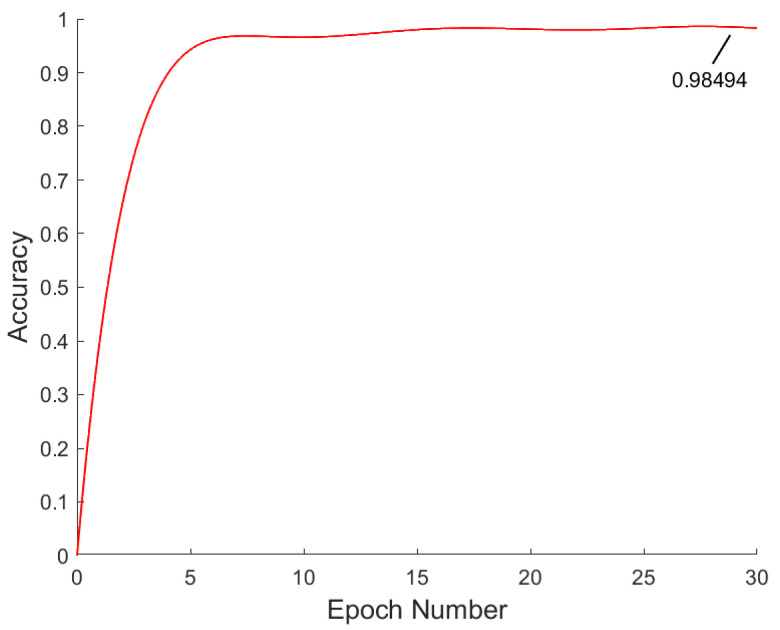
GazeNet training process.

**Figure 21 sensors-23-06239-f021:**
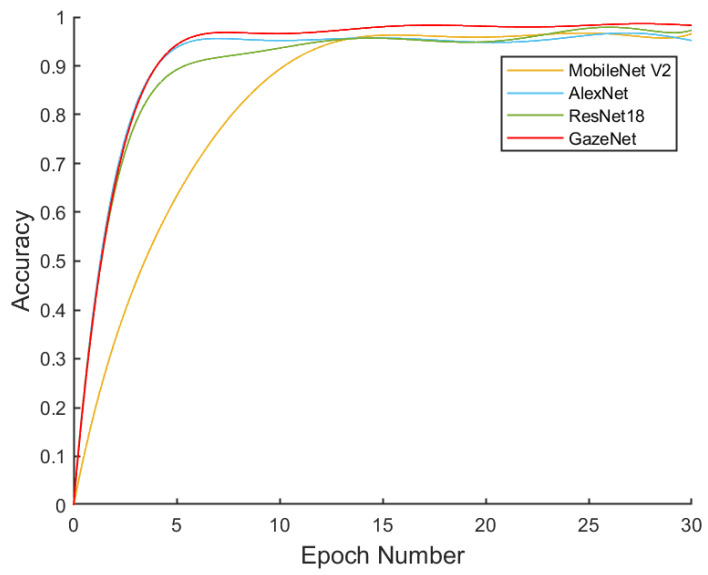
Training models for four different networks.

**Figure 22 sensors-23-06239-f022:**
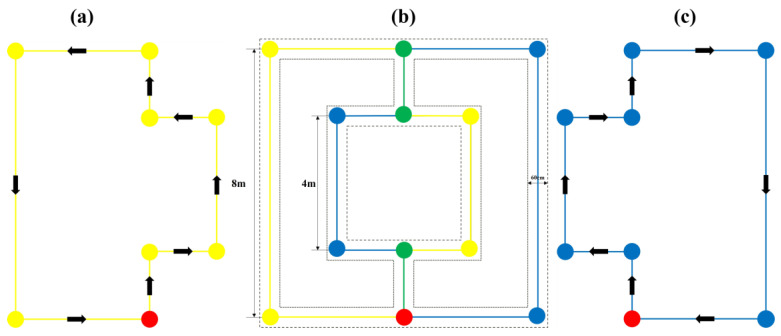
Experimental circuit diagram. (**a**) First lap, (**b**) actual route, and (**c**) second lap.

**Figure 23 sensors-23-06239-f023:**
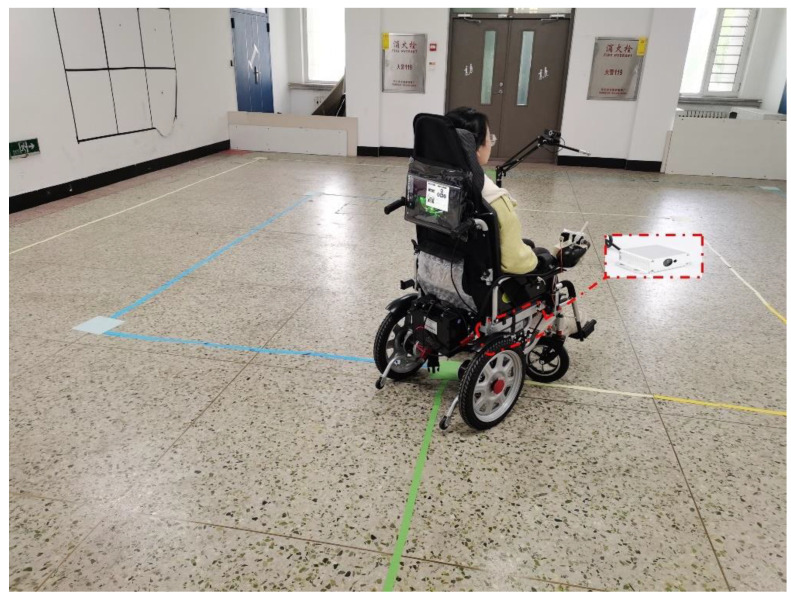
Actual test diagram.

**Figure 24 sensors-23-06239-f024:**
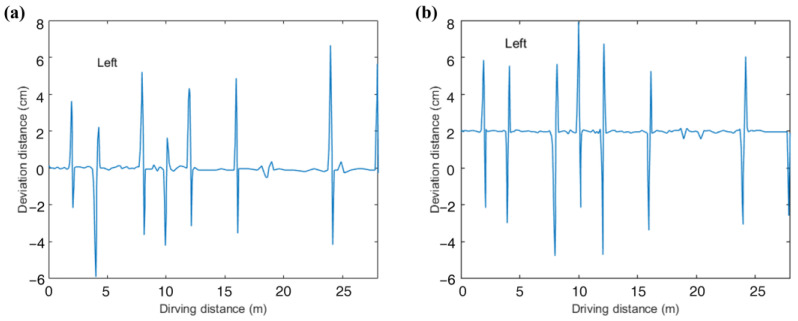
Variation of deviation with distance travelled: (**a**) lap 1 and (**b**) lap 2.

**Figure 25 sensors-23-06239-f025:**
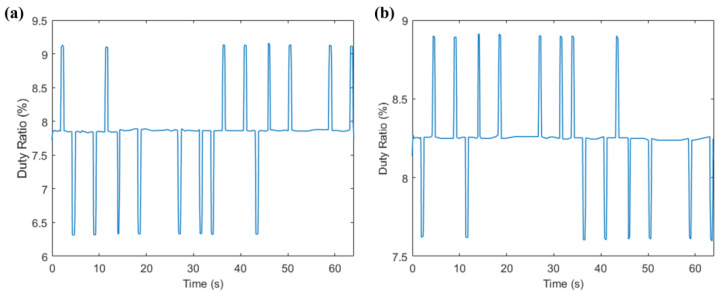
Servo duty cycle variation curve with motion time: (**a**) servo No. 1 and (**b**) servo No. 2.

**Table 1 sensors-23-06239-t001:** Comparison of the experimental results of different networks.

Model	Accuracy	Parameter Amount
GazeNet	98.49%	125,749
AlexNet	96.5%	564,003
ResNet18	97.5%	11,178,051
MobileNet-V2	96.9%	2,227,715

## Data Availability

The research data is available at: https://github.com/crzzx1/dataset (accessed on 1 May 2023).

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
