# Peer review of "Eye-Gaze Controlled Wheelchair Based on Deep Learning"

_sensors, 2023, doi:10.3390/s23136239_

Round 1

Reviewer 1 Report

The authors should add a related work (sub/)section where they should mention some limitations in the current literature the authors are trying to fix.

Dataset of the experiment could be available.

There are certain shortcomings in the description of the model and the analysis of the results should be improved.

-

Author Response

Dear Reviewer,

Thank you for your valuable comments on the revisions to our article. We have revised each of them in response to your review comments. Firstly, we commissioned MDPI to edit my manuscript in English. Secondly, we respect your idea to add a section of related work. And we have uploaded part of the dataset. Finally, we improved the variable annotations in the model description as well as the formulas, and explained some settings in more detail. We would like to thank the referee again for taking the time to review our manuscript. Please see the attachment.

Reviewer 2 Report

English language fine. Minor editing required.

Author Response

Dear Reviewer,

Thank you for your valuable comments on the revisions to my article. We have revised each of them in response to your review comments.

1.In Line 149, you stated “qualitative to quantitative", which is confusing. Data collection is a quantitative process, then what do you mean by the word "qualitative”?

We realized that there was a problem with the previous description from qualitative to quantitative when the dataset was built, so I deleted this confusing statement.

  1. In Line 188,what interference can the distance between the eyes have on human eye information. I think you can state more details about it.

We added content specifying what between the two eyes interferes with human eye information feature extraction.

  1. Line 320 You claimed that C denoted the number of channels in Line 306. Does S also denote the number of channels?

There is indeed an ambiguity in our previous description of C and S. The fact is that C stands for channel, while S stands for space.

  1. Line 337 In the formula, is the vector (x1, x2, x3,x4)T correctly written? And what does this vector mean here?

The previous vector (x1, x2, x3, x4)T (T is the superscript for transpose) is an example that really does not match the reality of this paper, and I have changed it to the specific feature vector (x1, x2...x15, x16)T (T is the superscript for transpose) in this paper. This vector represents here the 1D column vector obtained by spreading the output of the feature extraction network.

  1. Line 473.I think the eye in Figure 18 is right eye other than left eye.

You are right, we have corrected the error in the picture. Replaced it with a photo of the right eye.

  1. In the experiment part, you have considered the influence of blinks, however, there are also saccades that may influence the eye-tracking process. You can make an explanation.

The effect of sweeping you mentioned is of great significance to improve the utility of the wheelchair, and we have added the analysis of the effect of sweeping in the text and modified the flow chart accordingly.

Besides, We have commissioned MDPI to edit our manuscript in English. And We would like to thank the referee again for taking the time to review our manuscript.

Round 2

Reviewer 1 Report

The paper could be accepted in the current form

-